# Recent Advances in Aptamer Sensors

**DOI:** 10.3390/s21030979

**Published:** 2021-02-02

**Authors:** Samy M. Shaban, Dong-Hwan Kim

**Affiliations:** 1School of Chemical Engineering, Sungkyunkwan University (SKKU), Suwon 16419, Korea; samyshaban@skku.edu; 2Biomedical Institute for Convergence at SKKU (BICS), Sungkyunkwan University (SKKU), Suwon 16419, Korea; 3Petrochemicals Department, Egyptian Petroleum Research Institute, Cairo 11727, Egypt

**Keywords:** aptamer, colorimetric aptasensor, fluorometric aptasensor, electrochemical aptasensor

## Abstract

Recently, aptamers have attracted attention in the biosensing field as signal recognition elements because of their high binding affinity toward specific targets such as proteins, cells, small molecules, and even metal ions, antibodies for which are difficult to obtain. Aptamers are single oligonucleotides generated by in vitro selection mechanisms via the systematic evolution of ligand exponential enrichment (SELEX) process. In addition to their high binding affinity, aptamers can be easily functionalized and engineered, providing several signaling modes such as colorimetric, fluorometric, and electrochemical, in what are known as aptasensors. In this review, recent advances in aptasensors as powerful biosensor probes that could be used in different fields, including environmental monitoring, clinical diagnosis, and drug monitoring, are described. Advances in aptamer-based colorimetric, fluorometric, and electrochemical aptasensing with their advantages and disadvantages are summarized and critically discussed. Additionally, future prospects are pointed out to facilitate the development of aptasensor technology for different targets.

## 1. Introduction

Owing to the advances in various areas of life, the population has become more exposed to pollutants, whether from industry, agriculture, or medical waste, resulting in an increase in the pollution associated with humans’ daily lives, leading to an increase in the rate of diseases. In this context, the development of fast, simple, low-cost, high-sensitivity, and specific sensors for detecting pollutants or early stages of diseases is important.

Because of the great advances in molecular biology and genetic engineering, the use of RNA and DNA has expanded not only in biology, for storing and transmitting genetic information, but also in the identification of antibiotics, proteins, peptides, amino acids, and even small molecules. Gold et al. and Szostak et al. reported the first studies involving a specific targeting affinity toward proteins [1,2].

Aptamers consist of 3D-folded structures of single-stranded oligonucleotides with lengths of usually 20–60 bases of nucleotides selected in vitro via the systematic evolution of ligand exponential enrichment (SELEX) process. The SELEX process is applicable for single targets, complex target structures, or even mixtures without proper knowledge of their composition. SELEX can be used to select aptamers with high affinities and specificities for their targets and low dissociation-constant values, across the low nanomolar to picomolar range. Aptamers can be selected through the in vitro process independent of animals or cell lines. The selection of aptamers for toxic target molecules or molecules with no or low immunogenicity is possible. Different modifications can be introduced in the basic SELEX process for the selection of the desired aptamer specifications for a specific application. The aptamers can be functional for native conformations of target molecules on live cells, so cell surface transmembrane proteins can be considered as targets [3,4,5,6].

The 3D folded structure permits the formation of stable complexes with various targets, such as proteins, nucleic acids, and small molecules [7,8,9,10,11,12,13]. Compared to antibodies, these aptamers are characterized by numerous advantages such as a wide range of targeted molecules (inorganics, organics, cells, viruses, and bacteria, among others), facile preparation on a massive scale, low molecular weights, high temperature and pH stability, easier modification, and long-term storage stability. These advantages promote their importance and application in many fields such as biosensing, therapy, and diagnostics [14,15,16,17,18,19,20,21,22,23].

Because of their size-dependent physical and chemical properties, biocompatibility, electronic properties, magnetic properties, and ability to manifest biological signaling and transduction mechanisms, the nanomaterials are providing great advances in the field of sensors. Several nanomaterials have been synthesized with different sizes and shapes such as gold nanoparticles (AuNPs), silica nanoparticles, silver nanoparticles (AgNPs), magnetic nanoparticles (MNPs), and carbon quantum dots (CQDs) as unique templates for many applications such as biomaterial assays; the diagnosis, monitoring, and treatment of disease; and drug delivery [24,25,26,27,28,29,30,31,32,33,34,35,36,37,38]. Interestingly, nanomaterials are being used as potential candidates for immobilization using aptamers to obtain nanosensor probes for several targets, with more amplification and various signals such as colorimetric, fluorometric, electrochemical, or optical [31,39,40,41,42,43,44,45,46,47,48,49]. Sensor fabrication mainly depends on target recognition and signal transduction. Thus, nanomaterials provide desirable signal transduction for converting molecular recognition events into physically detectable fluorescent, colorimetric, and electrochemical signals [50,51]. Simultaneously, nanomaterials play a significant role regarding an increase in the immobilization density and orientation of aptamers due to the high surface areas of nanomaterials and, thus, provide high binding capacity for targets [52,53,54]. However, the high accumulation leads to the restriction of the 3D-structure formation. Because of that, the aptamer density on the nanomaterial surface needs to be precisely optimized [55,56].

This systematic review is focused on recent approaches to aptamer engineering for biorecognition for several objectives. Herein, previous and current advances related to aptamer-based sensing protocols are provided, highlighting the possible detected signals, with a focus on the use of different nanomaterials with distinct configurations. The most significant studies on aptasensor development have been collected, providing the possible strategies available for using aptasensors. Moreover, a deep discussion on colorimetric, fluorescence, and electrochemical detection strategies is provided.

## 2. Colorimetric Aptasensor

Among the available assay recognition signals, colorimetric methods are considered simple and efficient with a great potential for point-of-care diagnostics, as the detection responses are simply visually discerned by the naked eye using simple, low-cost, and effective instrumental techniques. The basic colorimetric strategies involve the detection of the target by a color change through the naked eye and simple instrumentation. Colorimetric biosensors based on aptamers have demonstrated their sensitivity and selectivity, in addition to their effective potential for rapid onsite diagnosis without complicated instrumentation. However, the colorimetric aptasensor has some limitations such as the influence of color from samples; the time-consuming nature of the fabrication process [57,58]; the difficulty in employing it in multiple-target assays, which are highly demanded in clinical diagnosis [59]; and the small range of optimized pH solutions [60].

Nobel metal nanomaterials, in particular, AuNPs and AgNPs, are excellent signal transducers for colorimetric analysis due to their significant optical properties associated with their particle size, size distribution, and shape [61,62,63]. The intrinsic surface plasmon resonance (SPR) properties of AuNPs and AgNPs contribute significantly to colorimetric signal generation [64]. AuNPs are used for color-change-based detection, in which the AuNPs are employed as nanoassembly units for immobilization with aptamers for constructing a colorimetric aptasensor because of their unique features, including simple synthesis and unique optical, thermal conductivity, and electronic properties [65,66,67,68]. Moreover, AuNPs show excellent biocompatibility compared to AgNPs, which are known for their toxicity [69]. Furthermore, the large specific surface area facilitates numerous adsorptions of biomacromolecules onto AuNP surfaces via electrostatic interaction, protecting them against aggregation and rendering them a good signal transducer for aptasensor construction [70,71]. Based on the unique colorimetric capabilities of the distance-dependent surface plasmon resonance of AuNPs, a great variety of label-free colorimetric bioassay strategies have been developed. The color of AuNPs is extremely sensitive to their dispersion and aggregation in a solution, comprising the interparticle plasmon coupling changing, inducing SPR shifts [72]. However, there are some limitations in using a colorimetric sensor based on the SPR response other than the common disadvantages of colorimetric sensors previously mentioned, including the fact that the SPR sensors are prone to interference due to not only the absence of a response to a change in the refractive index, but also non-specific binding that can induce interference [73,74], leading to false results and limiting their applicability. In some cases, variations in the plasmonic response depending on the location of the target on the surface of the nanoparticles [75,76] are detected. Moreover, large nanoparticle aggregates are unstable in solution, which could lead to some measurement error [77,78,79].

In this regard, the binding interaction induces a change in the refractive index around the surface of the AuNP that modulates its resonance angle. Therefore, the SPR has been used in an aptamer-based colorimetric strategy for the assay of several targets [80,81,82,83]. Gupta et al. proposed fast, easy, and cost-effective naked-eye visible detection for an *Escherichia coli* (*E. coli*) assay based on AuNPs’ aggregation as described in Figure 1A. In this assay, the AuNPs were covered with graphene oxide (GO), followed by functionalization with a specific aptamer for *E. coli*, establishing a stabilizing layer around the AuNPs. In the presence of the *E. coli* target, an aptamer–*E. coli* competitive interaction with AuNPs occurs, inducing the aggregation of AuNPs, with a distinct color change from red to blue and a low limit of detection (LOD) of 10^1^ cells/mL, which is observed visually by the naked eye [84]. Moreover, Lai et al. assayed malathion via salt-induced AuNP aggregation with aptamer assistance for improving the sensor selectivity, where the designated aptamer for malathion was initially adsorbed on the surface of the AuNPs, enhancing their colloidal stability against aggregation in a highly saline solution. In the presence of malathion, the specific aptamer departs from the AuNP surface to bind with the malathion target, leaving the AuNP surface unprotected and tending to aggregate in a highly concentrated NaCl solution, with a distinct color change from red to blue. For more signal amplification, Lai et al. used the aggregated AuNPs as a promoter for catalyzing the Fehling reaction, producing different-size Cu_2_O with different resonance scattering intensities and a LOD of 5.24 ng/L (15.86 pM) [85]. AuNPs were functionalized with aptamers equipped with colorimetric smartphone technology for a sensitive and selective Cd^2+^ detection strategy based on the AuNP aggregation induced by polydiene dimethyl ammonium chloride (PDDA) [86], in which the selective aptamer for Cd^2+^ shows a stable complementary hybridization with the PDDA. In the presence of Cd^2+^, the aptamer leaves the PDDA and binds to the Cd^2+^; AuNP aggregation is induced by the aptamer leaving the PDDA, which induces the solution’s color change from red to blue, with a linear range of 1–400 ng/mL and a LOD of 1 ng/mL, using a smartphone device as indicated in Figure 1B. Chen et al. also used aptamer–AuN-induced aggregation under the effect of a highly saline solution for assaying viable *Salmonella typhimurium* (*S. typhimurium*) bacteria using a dual aptamer approach, in which one was functionalized with magnetic beads to capture *S. typhimurium* and the other was used for PCR amplification, as shown in Figure 1C [87]. Finally, the hybridization between the amplicons and the AuNP probe took place for 5 min before adding MgSO_4_ to induce the aggregation. The system showed a linear relationship for *S. typhimurium* from 3.3 × 10^1^ to 3.3 × 10^6^ CFU/mL, with LODs of 33 and 95 CFU/mL for detection by the naked eye in pure culture and milk, respectively (Table 1) [87]. *S. typhimurium* bacteria have also been detected via AuNP aggregation through competitive aptamer–target binding under the influence of salts, as inducers of aggregation, with a linear range of 100 to 10^9^ CFU/mL and LOD of 16 CFU/mL, as reported by Jiecan [88]. The AuNP salt-induced aggregation mechanism was determined in an assay for aflatoxin M_1_ (AFM1) as described by Seyed et al. [89]. In this assay, a colorimetric aptasensor for the AFM1 assay was based on the protection of AuNPs against NaCl-induced aggregation via detaching the complementary strand of the aptamer (CS) from aptamer-modified streptavidin-coated silica nanoparticles (SNPs) in the presence of AFM1, as outlined in Figure 1D, with a linear dynamic range of 300–75,000 ng/L and LOD of 30 ng/L [89]. Similarly, a colorimetric aptasensor based on the aggregation of AuNPs as a signal transducer using cationic perylene probe as gold aggregating agent (CPP) was also provided by Lerdsrias et al. for assaying aflatoxin B1 (AFB1), with a LOD of 0.18 ng/mL [90], as shown in Figure 2A.

As previously discussed, several colorimetric aptasensors were developed based on AuNP-induced aggregation with different targets such as malathion, Cd^2+^, *Salmonella typhimurium*, and AFB1, changing the specific aptamer functionalized with the AuNPs [85,86,87,88,90]. The question is what the role of these targets and others in the stability and instability of AuNPs in solution is, and if this could influence the salt-induced AuNP aggregation or not, affecting the reliability of the proposed strategies.

Recently, Zong et al., and Liu et al. [91,92] proposed a study on the competitive binding affinities of both the aptamer and target toward the AuNPs and the effect of this binding on the sensing strategy based on salt-induced aggregation, as displayed in Figure 2B,C. Liu et al. investigated the adsorption affinity of dopamine as a target and its aptamer on the AuNP surface in an aptamer-based sensing strategy, revealing competition between the aptamer and target on the AuNP surface. To confirm the competition, a proposed competition between two different targets (i.e., melamine and K^+^) was also evaluated using their specific aptamers. The study aimed to examine if the binding between the target and AuNPs (dopamine and melamine were used as examples for strong target/AuNP interactions, while K^+^ was used for weak interactions) could affect the analytical reliability of the sensor. Liu et al. claimed two possible mechanisms, in which a aptamer specific for dopamine could adsorb on the surface of AuNP as described in Figure 2(Ba), or the dopamine target had a higher affinity to AuNPs than the aptamer, as shown in Figure 2(Bb). This study reveals that both the aptamer and dopamine have a strong affinity to the AuNP surface, and the dopamine itself induces different degrees of aggregation based on its concentration. A more interesting finding is that the dopamine adsorption on the AuNPs could inhibit the aptamer adsorption, revealing that the most probable mechanism is the one described in Figure 2(Bb). Finally, it was disclosed that the system based on AuNP aggregation-assisting aptamer is strongly kinetically controlled depending on the degree of the binding affinity and competition between the AuNP and both the target and aptamer and, also, depending on whether this target can also induce the stabilization or destabilization of AuNPs [91]. Based on this study, the free target in solution could cause problems for the sensor reliability, as also proposed by Zong et al. [92] in their study on the assay of As^+3^, as shown in Figure 2C. Therefore, the use of a stronger capping agent on AuNPs (cationic AuNPs or coating with nanomaterials such as graphene) that cannot be displaced compared to citrate could decrease the target’s strength of adsorption to the AuNP surface and avoid these limitations [93,94,95,96].

**Figure 2 sensors-21-00979-f002:**
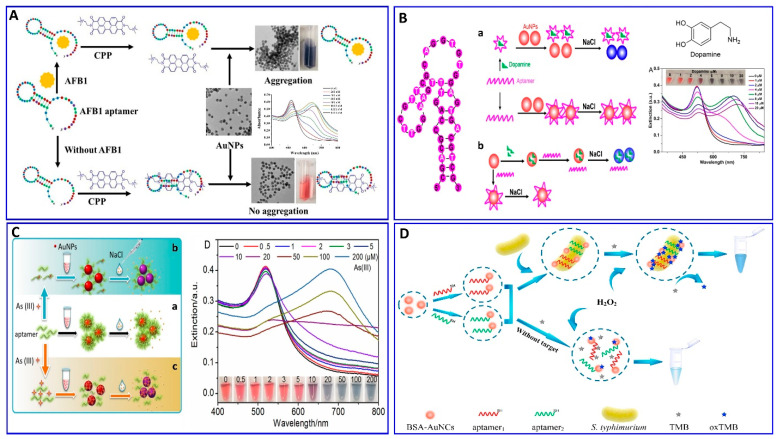
(**A**) Schematic illustration of AFB1 assay with the label-free colorimetric aptasensor using the competition between the specific aptamer, CPP, and AuNPs inducing AuNP aggregation, reproduced from [90]. (**B**) Schematic protocol of the dopamine assay via aptamer-assisted AuNP-induced NaCl salt aggregation with two possible mechanisms of sensing, reproduced from [91]. (**C**) Schemes of the As (III) assay based on the aptamer–As (III) competitive interaction inducing salt–AuNP aggregation, reproduced from [92]. (**D**) Schematic principle of *S. typhimurium* assay based on the peroxidase-mimicking activity of dual aptamers@BSA-AuNCs probes inducing a blue color for 3,3′,5,5′-tetramethylbenzidine (TMB), reproduced from [97].

The AuNPs were confined for using in a colorimetric aptasensor with different strategies other than aggregation, depending on the catalytic performance of the NPs and anisotropic growth of the AuNPs. Chen et al. proposed a colorimetric aptasensor for the selective and sensitive detection of *Salmonella typhimurium* (*S. typhimurium*) based on *S. typhimurium*’s affinity, to enhance the peroxidase-like activity of bovine serum albumin-stabilized gold nanoclusters modified with dual aptamers (aptamers@BSA-AuNCs), as outlined in Figure 2D [97]. In the presence of *S. typhimurium*, the aptamers@BSA-AuNCs could oxidize colorless 3,3′,5,5′-tetramethylbenzidine (TMB), producing the blue oxidized form, with a wide linear response to *S. typhimurium* in the concentration range of 10^1^–10^6^ CFU/mL and LOD of 1 CFU/mL [97].

Wang et al. proposed a colorimetric strategy for abscisic acid (ABA) detection based on the anisotropic growth of AuNPs in the presence of aptamer on the surface of the seed AuNPs [98]. In the presence of ABA, the functionalized aptamer on the surface of the AuNPs changed into a G-quadruplex-like structure and bound with ABA, triggering the departure of the aptamer from the surface of the AuNPs. In the presence of both the growth-promoting HAuCl_4_ and NH_2_OH as reducing agents, a uniform or anisotropic growth of AuNPs with a distinct change in the localized surface plasmon resonance (LSPR) could occur based on the amount of bound aptamer on the surface of the bare seed AuNPs, which is related to the concentration of ABA, with a linear relationship for 1 nM to 10 μM and LOD of 0.51 nM.

Furthermore, Seongjae Jo et al. and Wonjung Lee et al. [99,100] described a fabricated solid aptamer sensor for assaying cortisol and thrombin, respectively, by immobilizing the a specific designated aptamer with a specific sequence outlined in the (Table 1) on a glass substrate, as shown in Figure 3A,B. The changes in the LSPR, in the presence of cortisol and thrombin, showed linear ranges of 0.1–1000 nM and 0–10 μg/mL, with LODs of 0.1 nM and 1.33 μg/mL, respectively.

A colorimetric strategy other than AuNP aggregation was developed for a Cd^2+^ assay assisted by a specific Cd^2+^ aptamer based on the enzyme-mimicking activity of MoS_2_ nanocomposites, as reported by Tao et al. [101] (Figure 4A). The biotinylated Cd^2+^ aptamer was immobilized on the bottom of the microplate via biotin–avidin affinity; simultaneously, a functionalized complementary strand CsDNA–Au–MoS_2_ was used as a signal transducer, depending on its peroxidase-like activity toward the chromogenic TMB. This system exhibited a linear range of 1–500 ng/mL and a LOD of 0.7 ng/mL.

Magnetic nanoparticles (MNPs) have also been used as a nanoassembling template for bioassay targeting, either alone or in combination with other nanostructures, where they can be easily isolated from the solution via an external magnetic field [39,102,103,104,105]. Miao et al. reported a sensitive pH-based colorimetric strategy based on glucose oxidase (GOD) enrichment and catalysis for the amplified detection of cancer biomarkers with isothermal DNA signal amplification and multibranched rolling circle amplification (mb-RCA) [103]. The mb-RCA is integrated with a magnetic separation technology to avoid potential interference, as shown in Figure 4B; the system successfully detected the carcinogenic human PDGF-BB biomarker with a LOD of 0.94 pM [103]. The Fe_3_O_4_ nanoparticles incorporated with aptamers were used for assaying trace levels of adenosine triphosphate (ATP) biomarkers colorimetrically, as provided by Li et al. in Figure 4C [106]. The catalytic activity of the Fe_3_O_4_ nanoparticles in the conversion of the colorless TMB into blue TMB was enhanced significantly after functionalization with an ATP aptamer. In the presence of an ATP target, the target binds with its aptamer, which departs from the surface of the Fe_3_O_4_, decreasing its activity, with a detection range of 0.50–100 μM and a LOD of 0.09 μM. The toxic Pb^2+^ was also assayed via a colorimetric aptasensor based on the peroxidase-like activity of graphene/Fe_3_O_4_-AuNP composites, showing a linear range of 1–300 ng/mL and a LOD of 0.63 ng/mL, as reported by Tao et al. [107] (Figure 4D).

**Figure 4 sensors-21-00979-f004:**
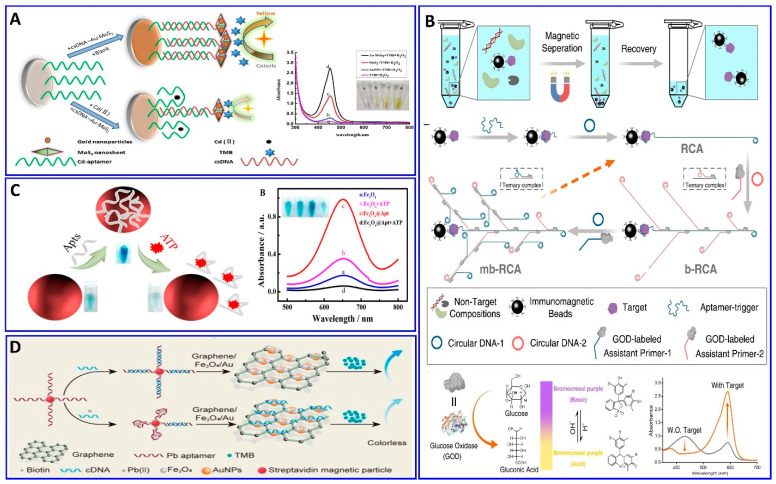
(**A**) Principal strategy for Cd^2+^ assay via the enhanced peroxidase-like activity of Au–MoS_2_-based aptamer, reproduced from [101]. (**B**) Principal protocol for the PDGF-BB assay via a glucose oxidase enzyme, inducing changes in pH-equipped immunomagnetic beads and mb-RCA as a signal amplifier, reproduced from [103]. (**C**) Schematic diagram showing the procedure and mechanism of the ATP assay via colorimetric aptamer technology based on the peroxidase-like catalysis properties of Fe_3_O_4_, reproduced from [106]. (**D**) Colorimetric procedure for Pb^2+^ assay based on the peroxidase-mimicking activity of graphene/Fe_3_O_4_-AuNPs aptamer technology, reproduced from [107].

Replacing the antibodies with aptamers in the ELISA protocol has attracted attention for assaying food-borne pathogens (e.g., *E. coli*), as reported by Duan et al.’s group [108]. First, the biotin aptamer was immobilized on the surface of an avidin-coated microplate to capture the *E. coli* bacteria. The detection aptamer was fabricated with Cu-MOF NPs through avidin–biotin affinity as a transducer signaling based on the peroxidase activity of the Cu-MOF NPs catalyzing the conversion of the colorless TMB into the oxidized blue structure and, then, yellow in the presence of the acid. This modified ELISA succeeded in detecting 16 to 1.6 × 10^6^ CFU/mL, with a LOD of 2 CFU/mL [108]. The introduction of the aptamer technology in the sandwich ELISA was used in the detection of 17*β*-estradiol-E2 (17*β*-E2) by Huang et al. [109] (Figure 5A). Xing et al. reported the detection of toxins, AFB1, ochratoxin A, and zearalenone, with a green ELISA based on a single-stranded-binding protein (SSB)-assisted aptamer with horseradish peroxidase (HRP) technology to produce the signal as demonstrated in Figure 5B [110]. This modified ELISA aptamer showed low limits of detection in corn: 112, 319, and 377 ng/L for AFB1, ochratoxin A, and zearalenone, respectively. Wei et al. described a multicolor and photothermal assay of prostate-specific antigen (PSA) as a dual signals based on ELISA modified aptamers, as outlined in Figure 5C [111]. In the modified ELISA, an Fe_3_O_4_ NP–graphene oxide composite was functionalized with complementary DNA and served as the signal producer for assaying the PSA. In the absence of PSA, the Fe_3_O_4_ NP–graphene oxide–DNA probes were captured by the PSA aptamer; therefore, the Fe_3_O_4_ NPs were transformed into Prussian blue (PB) using a HCl solution, which was then mixed with potassium ferricyanide to produce a multicolor signal; simultaneously, the PB can produce a thermal signal upon near-infrared laser irradiation, with LODs of 0.15 and 0.31 ng/mL for colorimetric and photothermal assays.

Li et al. introduced a simple, eye-based microfluidic aptasensor (EA-Sensor) for the rapid assay of *E.coli O157:H7* [112], as described in Figure 5D. Coupling between the aptamer and hybridization chain reaction (HCR) amplification was considered based on distance visualized readout technology for quantitative assays (Figure 5D). The HCR amplified the signal 100-fold after the *E.coli O157:H7* was recognized by the aptamers. The system exhibited good linearity in the range of 500–5 × 10^7^ CFU/mL and a LOD of 250 CFU/mL.

## 3. Fluorescence Aptasensor

Fluorescence-based aptasensors are characterized by high sensitivity, large-detection ranges, multiplexing capabilities, rapid assaying, and the highly selective recognition of aptamers for several targets, which can be distinguished under UV light, in comparison with the colorimetric sensor’s signals that are distinguished using visible light. The integration of both fluorescent materials (fluorophore dyes and fluorescent nanoparticles such as upconversion nanoparticles (UCNPs), GO, and CQDs) and aptamers can produce high sensitivity and selectivity, and a rapid analysis strategy, making them useful candidates for fluorescence-aptasensor bioassays [113,114,115]. The recognition affinity between aptamers and analytes induces conformational changes in the aptamer. This process can trigger changes in the fluorescent emission properties of the fluorophore dye or fluorescent nanomaterials, owing to changes in the original environments of these materials.

The design of fluorescent aptasensors requires the use of hairpin aptamers (aptabeacons), which are labeled with either a fluorophore or a quencher. Forster resonance energy transfer (FRET) typically uses a donor fluorophore and an acceptor quencher material. Different strategies are often employed via FRET operations, which are performed through either “signal-on” or “signal-off” assaying protocols. These operations are based on the disparity in the fluorescence responses of the fluorophores as a function of the potential and unique aptamer–target binding and the conformational-change degree [116,117,118]. Owing to the conformational change of the aptamer induced via target interactions, the probe was successfully switched on/off via the FRET mechanism.

Some of the relevant fluorescent strategies were employed for the assaying of several targets based on integrated aptamers and fluorescent materials as described in the following Figure 6, Figure 7 and Figure 8. Khan et al. provided a colorimetric and fluorometric method for the assaying of T-2 (trichothecenes A) with the assistance of aptamers [119], as indicated in Figure 6A. In this method, through the FRET mechanism, a green-emitting L-arginine@6-aza-2-thiothymine-protected gold nanocluster capped with polyacrylic acid (PAA@Arg@ATT-AuNCs) was quenched via dispersed AuNPs in the absence of the T-2 target. In the presence of the target, the PDDA became free in the solution and induced the aggregation of the AuNPs that were unable to quench the fluorescence probe (LOD: 0.57 pg/mL, and linear range: 0.001−100 ng/mL) [119].

Some studies have focused on detecting several targets based on the aptamer-competitive quenching of carbon quantum dots and graphene materials [120,121,122,123,124,125]. Recently, GO was used as a “nanoquencher” for fluorescent materials via the FRET strategy with ssDNA as a nanoscaffold, capable of adsorbing to the GO surface through π–π stacking interactions with the nucleotide bases [126,127,128,129]. Shirania et al. proposed a fluorometric aptasensor for assaying digoxin (DGX) based on the quenching effect of an aptamer/gold nanoparticle probe on the fluorescence intensity of graphitic carbon nitride nanosheets. The FRET strategy was employed, and the presence of DGX resulted in fluorescence intensity restoration (Figure 6B), with concentrations ranging from 10 to 500 ng/L and a LOD of 3.2 ng/L [120]. Tan et al. [121] proposed another protocol system that uses a ratiometric fluorescent nanoprobe, in which dual emission at 518 and 608 nm is induced by a single excitation. This system was applied for the assaying of zearalenone (ZEN), which could quench green-emitting g-QDs linked to an aptamer via an electron transfer mechanism (Figure 6C), with a LOD of 7.5 nM [121]. Khan et al. proposed that the water-soluble mycotoxin patulin (PAT) produced by several fungal species, including from the genera *Aspergillus*, *Penicillium*, and *Byssochlamys*, be assayed via fluorometric aptasensors (Figure 6D), where the carboxyfluorescein (CFL) fluorescent dye tagged with a PAT aptamer was quenched by –COOH-functionalized multiwall carbon nanotubes (CA-MWCNTs). The presence of PAT induces the recovery of the fluorescence, with a LOD of 0.13 ng/mL [130].

Recently, Guo et al. reported that thrombin and ATP aptamers exhibited quenching effects on positively prepared polyethylenimine (PEI-CDs). In the presence of thrombin or ATP, the respective aptamer binds with its target, leading to the restoration of the fluorescence, with LODs of 1.2 and 13 nM for thrombin and ATP, respectively [95]. The aptamer has been functionalized with fluorescent dyes such as the TAMRA dye, as proposed for the assaying of AFB1 (Figure 6E) [123]. In the absence of AFB1, the specific aptamer labeled with TAMRA was adsorbed on the surface of the UiO-66-NH_2_ (metal–organic framework, MOF). This MOF can quench the fluorescence of the TAMRA with a sensitivity of 0–180 ng/mL and a LOD of 0.35 ng/mL. 3,4,9,10-perylenetetracarboxylic acid diimide (PTCDI), an affordable and low-cost fluorophore dye, was used for the assaying of ampicillin (AMP). This assaying was based on the competitive quenching between the AMP aptamer, and the complementary strand (CS) and AuNPs (Figure 6F), with a linear range of 100 to 1000 pM and LOD of 29.2 pM [131].

A method for signal amplification was introduced by Wen et al., who applied dual atom transfer radical polymerization (ATRP) technology for amplifying the fluorogenic signal during the assaying of gamma-interferon protein (IFN-*γ*). As outlined in Figure 6G [132], two aptamers were used to sandwich the protein, where the second aptamer was functionalized with azide, which, upon polymerization, produces an amplified fluorogenic signal (LOD: 0.178 fM).

Polydiacetylene (PDA) liposomes were immobilized with an aptamer-tagged fluorophore (Cy3–Apt) for assaying mucin 1 (MUC1), as indicated in the proposed scheme shown in Figure 6H, with a LOD of 0.8 nM [133], based on the spectral overlap of Cy3’s fluorescence emission and the absorbance of PDA–Apt liposomes [133]. Fan et al. synthesized Trich three-way junction (AT-TWJ) DNA-stabilized CuNPs as a fluorescent transducer and magnetic Fe_3_O_4_@GO as a single-stranded DNA (ssDNA) adsorbent for the detection of isocarbophos [134] (Figure 6I). The dsDNA is hybridized by the isocarbophos aptamer, while the complementary DNA plays a role as a recognition unit. In the presence of isocarbophos, the released dsDNA and the two ssDNA strands can assemble into AT-TWJ DNA. This facilitates the formation of AT-TWJ DNA template CuNPs with a strong fluorescence response toward isocarbophos (concentration range: 10–500 nM), with a LOD of 3.38 nM.

Carbon nanotubes (CNTs) are considered perfect nanoquenchers for fluorophore-labeled ssDNA, owing to the stacking of nucleotides on the surface of the multiwalled carbon nanotubes (MWCNTs), which induces quenching via the FRET system. The hybridization of the aptamer with the target leads to the restoration of the fluorescence. Therefore, the ssDNA–MWCNT assembled fluorescence probe was used for the assaying of Hg^2+^ [135].

**Figure 6 sensors-21-00979-f006:**
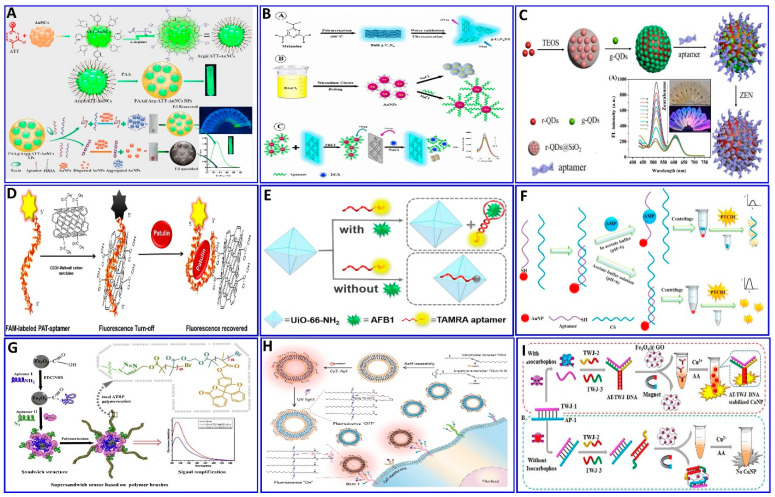
(**A**) Schematics showing the preparation of PAA@Arg@ATT-AuNC as a fluorescent probe for trichothecenes A (T-2) toxin assay, reproduced from [119]. (**B**) Graphical illustration for the assaying of digoxin (DGX) using the aptamer/AuNPs/g-C3N4NS sensor probe, reproduced from [120]. (**C**) Schematic illustration of the zearalenone assay using a ratiometric fluorescent nanoprobe with dual emission at 518 and 608 nm, reproduced from [121]. (**D**) Schematic description of the protocol employed for a patulin (PAT) assay via CA-MWCNTs) with quenching of aptamer-tagged carboxyfluorescein, reproduced from [130]. (**E**) Schematic description of AFB1 detection based on UiO-66-NH_2_ and aptamer-functionalized TAMRA dye, reproduced from [123]. (**F**) Schematic of the fluorescent aptasensor for an AMP assay based on the competitive quenching between the AMP aptamer complementary strand and AuNPs, reproduced from [131]. (**G**) Schematic showing an ultrasensitive fluorometric aptasensor for IFN-γ detection by dual atom transfer radical polymerization (ATRP) amplification, reproduced from [132]. (**H**) Schematic showing the fabrication of PDA–Apt liposomes and fluorescence imaging of plasma membrane glycoprotein MUC1, reproduced from [133]. (**I**) Schematic description of isocarbophos assay via fluorometric aptsensor based on AT-rich three-way junction DNA template copper nanoparticles and Fe_3_O_4_@GO, reproduced from [134].

Rare-earth-element-doped UCNPs can convert low-energy light into high-energy light with short wavelengths via the photon mechanism. UCNP materials are characterized by high quantum yields, narrow emission peaks, excellent photostability, long fluorescence lifetimes, environmental friendliness, low toxicity, bulky anti-Stokes shifts, and high resistance to photobleaching. Moreover, infrared light induces minimal photodamage to the biological targets without exciting the biological fluorophoric substance, which can be confined in the matrix [80,136,137,138,139]. UCNPs were used in the development of fluorescence aptasensors for assaying different targets [140,141,142,143,144]. However, they still have some limitations such as the difficulty in distinguishing different target concentrations by the naked eye upon excitation by a laser beam at 980 nm, where the target can be only be detected using instrumentation. However, UCNPs have been used for aptasensing development, in which a new biosensor for *E. coli* bacteria has been developed based on the FRET between aptamer-functionalized UCNPs as donors and layered tungsten disulfide (WS_2_) as the acceptor. This was achieved through the attachment of the aptamer to the WS_2_ surface via van der Waals forces. In the presence of *E. coli*, the specific aptamer binds preferentially to *E. coli*, causing a conformation of the aptamer that induces the dissociation of the probe away from the surface of the WS_2_ nanosheets. Therefore, part of the fluorescent intensity is restored as a function of *E. coli* (Figure 7A), with concentrations ranging from 85 to 85 × 10^7^ CFU/mL and a LOD of 17 CFU/mL [140]. Moreover, *E. coli* was detected using immobilization between UCNPs and aptamer technology as described in Figure 7B, in which the fluorescence emission intensity of the MNP–aptamer–cDNA–UCNP complex decreases upon increasing the *E. coli* concentration, with a linear system range of 58–58 × 10^6^ CFU/mL and LOD of 10 CFU/mL [141]. A similar protocol was employed for a malathion assay using UCNPs (Figure 7C), with a linear range of 0.01 to 1 μM and LOD of 1.42 nM [142]. Pb^2+^ was detected using aptamer-functionalized UCNPs incorporated with magnetic Fe_3_O_4_^−^ (MNPs), with modified AuNPs, with a detection range of 25–1400 nM and LOD of 5.7 nM [143]. Basically, the dispersion and aggregation affinity were harnessed to produce a quencher for the fluorescent materials, as used in the assaying of chlorpyrifos (CPF) [144]. Fluorescent terbium (III)-based MOFs (Tb-MOFs) were quenched by dispersed AuNPs, and the fluorescent intensity was restored by PDDA-aggregated AuNPs, owing to the binding between the aptamer and CPF, with a sensitivity of 5–600 nM and LOD of 3.8 nM [144]. Qiu et al. proposed a ratiometric biosensing fluorescent sensor for ATP based on multicolor silver nanoclusters (AgNCs; Figure 7D) [145]. Once the sensor is exposed to the ATP, the binding between the ATP and aptamer complexes opens the hairpin structure and releases the anchor sequence, which hybridizes with the ssDNA and induces the turning off of the green fluorescence, while the red fluorescence is turned on, with a LOD of 0.38 μM [145].

**Figure 7 sensors-21-00979-f007:**
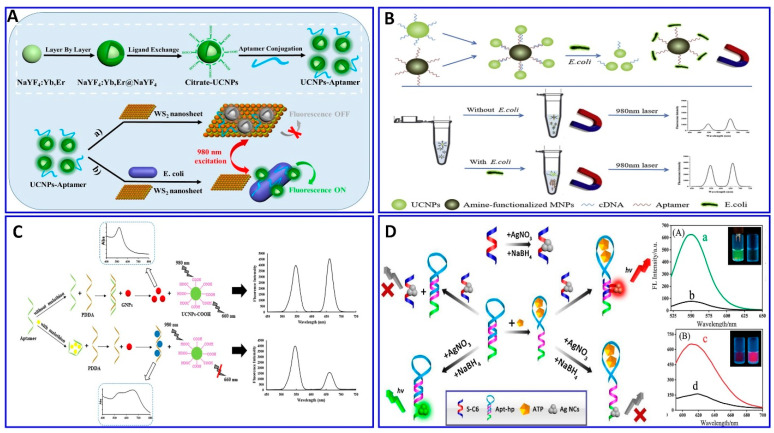
Schematic description of the (**A**) sensing platform for *E. coli* detection using fluorescent UCNP–WS2 nanosheet, reproduced from [140]. (**B**) Proposed fluorescence aptasensor for *E. coli.* using MNP–aptamer–cDNA–UCNP probe, reproduced from [141]. (**C**) Fluorometric aptasensor for malathion assay based on the FRET between UCNPs and GNPs, reproduced from [142]. (**D**) Proposed scheme of ATP assay based on ratiometric fluorescence from the binding between the ATP and aptamer complexes, reproduced from [145].

As mentioned above, aptamers are sequences of nucleic acid, and because of that, they can be integrated with a DNA system for fluorescent signal amplification to improve the detection limit and, therefore, be used in the early diagnosis of diseases. These amplification processes include rolling-cycle amplification, strand displacement reactions, hybridization chain reactions, and hairpin DNA cascade hybridization reactions [146,147,148,149,150,151]. Ning et al. used exonuclease III (Exo III)-assisted target-recycling amplification integrated with aptamer technology for the ultrasensitive assaying of adenosine triphosphate (ATP) via the quenching of the DNA/AgNCs fluorescent probe with gold nanorods (+) (AuNRs), with a LOD of 26 pM [152], as described in Figure 8A. The amplification of the signal using exonuclease I (Exo I) was also integrated with the fluorescent aptasensor for assaying AFB1 [153]. In the presence of AFB1, the aptamer structure changes, preventing its digestion by Exo I. Therefore, the addition of SYBR intensifies the fluorescence as a function of the AFB1 concentration, with a wide linear range of 2–400 ng/mL and LOD of 1.82 ng/mL [153]. Employing the same technology, the exonuclease integrated with an aptamer was used to assay acetamiprid [154]. In the absence of acetamiprid, the specific aptamer hybridizes with the complementary DNA labeled with ferrocene (cDNA–Fc), without a change in the fluorescence intensity after the addition of the exonuclease (RecJf), while in presence of acetamiprid, it will bind with the aptamer, which leaves the cDNA–Fc, and then digested by the RecJf exonuclease, liberating the Fc to interact with cyclodextrin and initiating photoinduced electron transfer, as outlined in Figure 8B [154].

Aggregation-induced emission (AIE) was conducted for assaying exosomal tumor-associated proteins using GO as a quencher and TPE-TA (tertiary amine-containing tetraphenylethene) as a fluorescent dye (Figure 8C). The positively charged TPE-TA binds one aptamer and aggregates, resulting in an amplified fluorescence. When the target exosomes are introduced, the aptamer preferentially binds with its target. Thus, the TPE-TA/aptamer complexes detach from the GO surface, inducing the “turning on” of the fluorescence, with a response range of 0.68–30.4 pM and a LOD of 0.57 pM [155].

The aptamers themselves can be modified with a fluorophore and quencher materials at both ends without affecting their binding affinity toward the targets, so a FRET aptasensor strategy can be easily constructed in the presence of targets inducing fluorescence quenching [156,157,158]. Kang et al. provided a dual-mode readout DNA biosensor combining both an aptamer and a DNAzyme to assay ATP with two different mechanisms, using fluorometric and colorimetric signals (Figure 8D), respectively. The presence of ATP induces attraction between the two strands of the aptamer, which induces the quenching of the fluorescence dye on one aptamer end via the quencher labeled on the other end of the aptamer. At the same time, the ATP induces the folding of the DNAzyme into a G-quadruplex, enhancing its peroxidase activity in oxidizing 2′-azino-bis(3-ethylbenzothiazoline-6- sulfonic acid) (ABTS). The system showed a large dynamic range, 1−500,000 μM, more than five orders larger than that reported to date for ATP assays [159]. Lan et al. used a strategy for assaying tetrodotoxin (TTX) based on the difference in the fluorescence response of berberine as a result of TTX–aptamer interaction with the analyte (Figure 8E), exhibiting a linear range of 0.1–500 nM and a LOD of 0.074 nM [160]. A molecular imprinting polymer (MIP) was used for assaying Cd^2+^ in environmental and agricultural samples with the assistance of a specific aptamer. The MIP was integrated with aptamers to make dual recognition units, while carbon-doped sulfur, nitrogen quantum dots, and AuNPs (SN-CQD/Au) were used as a fluorophore. The fluorescence was quenched via Cd^2+^, with a linear response range of 20 pM–12 nM and a LOD of 1.2 pM. [161].

Zhang et al. developed a simple, sensitive, and label-free method to assay the cyanotoxin microcystin–leucine–arginine (MC-LR), using fluorescent dsDNA–CuNCs as a fluorescent probe, comprising an aptamer for MC-LR and its complementary ssDNA. The presence of the MC-LR target results in the uptake of the aptamer from the dsDNA–CuNCs; therefore, the fluorescent signal exhibits a sensitivity range of 0.01 to 1000 μg/L and LOD of 4.8 ng/L [162]. Sharma et al. provided a technique for chloramphenicol analyte detection using three oligonucleotide complexes comprising an aptamer, fluorophore oligomer, and oligomer quencher as a sensor platform (Figure 8F), with a linear range of 10–107 pg/mL and a LOD of 0.987 pg/mL [163].

**Figure 8 sensors-21-00979-f008:**
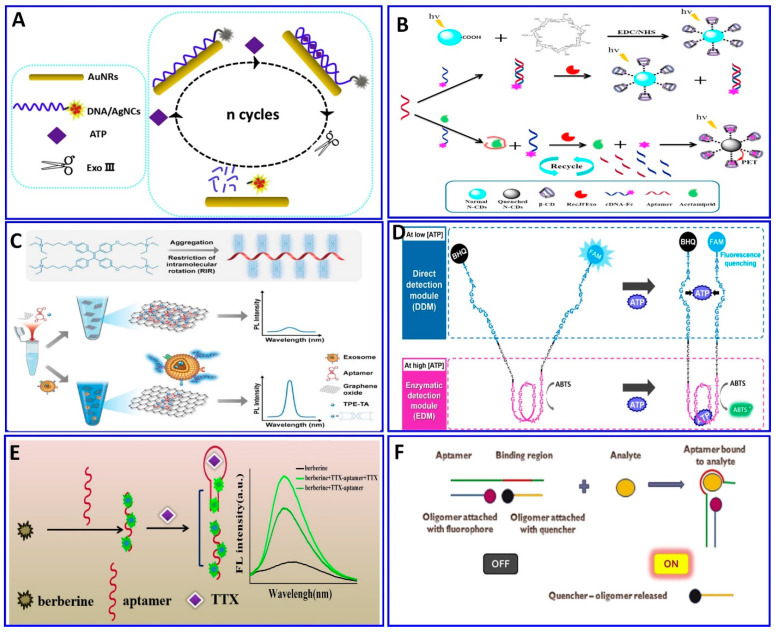
(**A**) Schematic illustration of the ATP assay via quenching of DNA/AgNCs by AuNRs functionalized with aptamer and Exo III for signal amplification, reproduced from [152]. (**B**) Fluorometric assay for acetamiprid using exonuclease integrated with the aptamer protocol, reproduced from [154]. (**C**) Exosomal assay based on the AIE mechanism using graphene oxide (GO) as a quencher for tertiary amine-containing tetraphenylethene (TPE-TA) dye, reproduced from [121]. (**D**) Dual-mode fluorescent aptasensor using both aptamer and a DNAzyme to assay ATP with two different mechanisms, fluorometric and colorimetric signals, reproduced from [159]. (**E**) TTX assay based on the difference in fluorescence response of the berberine reporter, reproduced from [160]. (**F**) Turn-on fluorescence aptasensor assay for chloramphenicol based on oligomer quencher release, reproduced from [163].

## 4. Electrochemical Aptasensor

An electrochemical aptasensor was fabricated using an aptamer as a bioreceptor and an electrochemical transducer, which translated the target–aptamer affinity into a measurable electrochemical signal through potentiometry, voltammetry, amperometry, impedimetry, or electrochemiluminescence. The potentiometric approach involves the measurement of the potential between the probe and the reference electrode without any net charge transfer. The amperometric approach is based on applying a potential and allowing a redox reaction to occur. The signal is defined as the current between the electrode and the counter-electrode. The voltammetric strategy includes the sweeping potential over time and recording the corresponding current; this strategy is based on a three-electrode system, with working, reference, and counter electrodes. The impedimetric technique involves measuring the charge transfer rate on the surface of the electrode for a kinetic study.

The electrochemical aptasensors were modified with various nanomaterials such as carbon-based nanomaterials, metal–organic frameworks (MOFs), AuNPs, and polymers for signal amplification [164,165,166,167]. The aptamers were immobilized on the electrode surface via π–π interactions, hydrogel grafts, biotin–avidin interactions, thiol–gold self-assembly, and carboxyl–amine covalent reactions [168,169,170,171,172,173].

The electrochemical aptasensor mainly depends on the interactions occurring on the surface of transducer as a result of the induced reaction between the target and its specific aptamer, providing amperometric or potentiometric electrochemical signals. Another technique is based on the increase in charge transfer resistance via the impedance technique [174]. Therefore, an electrochemical aptasensor was provided for the detection of several targets, such as ampicillin (AMP), avian influenza virus (H5N1), carbohydrate antigen 125, Pb^2+^, lysozymes, insulin, thrombin, CD44, vanillin, circulating human MDA-MB-231 breast cancer cells, bisphenol A, furaneol, and Hg^2+^, among others [8,46,175,176,177,178,179,180,181,182,183,184,185,186,187,188,189].

The chratoxin A (OTA) toxin was detected electrochemically, specifically using the OTA aptamer as demonstrated in Figure 9A. The probe was fabricated via the immobilization of the OTA aptamer on the gold layer, and the surface of the aminated polystyrene, which binds with the glassy carbon electrode (GCE) via peptide bonds, was coated with the aptamer functionalized with gold, exhibiting a sensitivity of 1 × 10^−5^ to 10 nM, and a LOD of 3.3 × 10^−3^ pM [190]. Amoxicillin was detected in wastewater via an ultrasensitive impedimetric aptasensor based on the synergetic effect of the surface of the GCE being coated with TiO_2_-g-C_3_N_4_ and AuNPs (Figure 9B). Because of the formation of the Au–S bond between the AuNPs and the modified thiol aptamer, competition on the surface of the GCE occurs between the amoxicillin and its aptamer, achieving a detection sensitivity of 0.5–3 nM and a LOD of 0.2 nM [191].

A sandwich-type electrochemical aptasensor has been fabricated for a thrombin assay, as demonstrated in Figure 9C. The sensor probe was fabricated on a conductive supramolecular polymer hydrogel (CSPH) modified electrode, on which the thrombin-binding aptamer 1 (TBA1) was immobilized via amide bonds, while the thrombin-binding aptamer 2 was modified using magnetic nanoparticles (MNP-TBA2) as signal amplification probes; the sandwich-type electrochemical aptasensor showed a linear range of 1 pM to 10 nM, with a LOD of 0.64 pM [192]. A sensitive electrochemical aptasensor for detecting prostate-specific antigen (PSA), based on coral-like polyaniline (PANI)/AuNPs and having peptides as antifouling materials to reduce nonspecific adsorption, was fabricated on a GCE surface (Figure 9D). A PSA aptamer was immobilized on the electrode for a PSA assay, with a wide linear range from 0.1 pg/mL to 100 ng/mL and a LOD of 0.085 pg/mL [169].

Recently, enzymes such as HRP, glucose oxidase, and alkaline phosphatase, and electroactive compounds such as QDs, ferrocene (Fc), ferrocyanide, methylene blue (MB), and Cd nanoparticles were successfully incorporated in electrochemical aptasensor technologies to be used as signal enhancers [114,194]. A different strategy was integrated in an electrochemical aptasensor for assaying different targets—AFB1-activated protein C, OTC, patulin (PAT), thrombin, and Pb^2+^, among others—by monitoring the electrochemical signals of the released labeling substrates as a result of competitive targets [13,43,195,196,197,198,199,200]. A portable electrochemical aptasensor was fabricated for assaying OTC using a AuNP/cMWCNT/cDNA@thionine probe as a signal-releasing tag in the presence of the target [196], as outlined in Figure 10A. In the presence of OTC, the AuNP/cMWCNT/cDNA@thionine tag was released from the surface of the gold electrode, inducing a change in the electrical signal, with a linear range of 1 × 10^−13^–1 × 10^−5^ g/mL and a LOD of 3.1 × 10^−14^ g/mL. The same technology was also used for assaying PAT; MOF@Methylene blue was used as the signal tag released from the electrode surface as a function of the PAT concentration, with a linear range of 5 × 10^−8^–5 × 10^−1^ μg/mL and a LOD of 1.46 × 10^−8^ μg/mL [43], as shown in Figure 10B. A homogeneous electrochemical aptasensor was fabricated based on the departure of a target-responsive label, the electroactive dye methylene blue (MB), from an aptamer-gated zeolitic imidazolate framework-8 (ZIF-8) as a function of the thrombin concentration (Figure 10C), with a linear range of 1 fM to 1 nM and LOD of 0.57 fM [13]. Employing the same technology, Pb^2+^ was detected with a range of 0.5–25 ppb and a LOD of 0.6 ppb, using toluidine blue as a redox-released tag signal liberated in the presence of Pb^2+^ [197]. Enzymatic catalysis was incorporated in an electrochemical aptasensor for signal amplification [201,202]. Fu et al. assayed H5N1 using the electrochemical aptasensor method integrated with enzymatic catalysis, as shown in Figure 10D [201]. Magnetic beads were fabricated with an H5N1 aptamer to enhance the selectivity, followed by immobilization with concanavalin A (ConA), GOx, and AuNPs for signal enhancement in the glucose-fluid-inducing formation of gluconic acid. The system showed a LOD of 8 × 10^–4^ HAU (hemagglutination units) in a 200 μL sample [201].

The electrochemical method was integrated with amplification strategies for assaying numerous targets, such as sulfadimethoxine, malathion, OTA, kanamycin, ampicillin, atrazine (ATZ), and acetamiprid [138,203,204,205,206,207,208,209,210]. Sulfadimethoxine was assessed electrochemically with the assistance of nuclease amplification (Figure 11A), in which a gold surface of an electrode was immobilized with a DNA probe, and then hybridized with a specific aptamer for sulfadimethoxine, forming a dsDNA that inhibited their digestion via nuclease P1. The sulfadimethoxine aptamer leaves the DNA probe in the dsDNA in the presence of sulfadimethoxine, allowing the digestion of the dsDNA via nuclease P1 and inducing the amplification of the electrochemical signal, with a response range of 0.1–500 nM and a LOD of 0.038 nM [203]. Malathion was detected with the assistance of exonuclease I (Exo I) as an electrochemical signal enhancer, which enhanced the signal by over two times, as shown in Figure 11B. Moreover, the electrodeposition synthesis of PDA–AuNPs provided satisfactory biocompatibility and electrical conductivity for the sensor. Exo I was used to promote the autocatalytic cycling of malathion by enhancing the current change, with a linear response range of 0.5–600 ng/L [204]. OTA and ATZ were detected electrochemically and photoelectrochemically with the assistance of cycling amplification, with a linear range of 0.01 to 1 ng/mL and a LOD of 0.004 ng/mL for OTA [205] and linear range of 50 fM to 0.3 nM with LOD of 12.0 fM for ATZ [206], as outlined in Figure 11C,D.

## 5. Conclusions and Future Outlook

The distinctive and impressive advantages of aptamers compared with antibodies permit them to be preferred in molecular diagnostics for a wide range of biomarkers. Aptamers have intrinsic advantages, such as their availability for both chemical modifications and conjugation with different labels, facilitating their ability to be used to construct a sensitive and highly selective platform sensor. In this review, recent advances in the different methods of employing aptasensors including colorimetric, fluorometric, and electrochemical strategies were discussed. The simplicity and small sizes of the aptamers combined with the versatile optical properties and large surface areas of nanomaterials leads to a platform with great potential for highly sensitive and selective biological recognition and signal transduction for various analytes such as metals, small molecules, toxins, proteins, cells, and bacteria with lower LODs and high sensitivity and selectivity.

In this review, each detection method with its own different strategies was emphasized briefly with schematic designs and all the aptamer information, including the detection ranges of the discussed aptasensors, summarized in Table 1. Each aptasensor method has advantages, but the limitations of each method have to be considered before assaying the biomarkers. The optical strategies are considered promising assay methods, as a sensitive response could be achieved; however, some limitations should be considered before assigning a suitable strategy. The colorimetric aptasensor sensors are considered more promising for point of case testing (POCT) owing to naked-eye readout; however, their sensitivity fails to meet the required criteria owing to the higher LODs compared to other methods. Moreover, some limitations were demonstrated in the AuNP–salt-induced aggregation, in which the surface of the AuNP was easily accessible by several targets and aptamers, affecting the reliability of the sensor. This hurdle was overcome by using a strong capping agent [93,94,96]. Despite the fact that the fluorescent aptasensor achieved a high sensitivity for the assay of different targets compared to colorimetric, it requires laboratories and clinical centers with infrastructure for achieving POCT diagnosis, and the limitations of the photobleaching of the fluorescent molecules over time, compromising their stability, is considered an obstacle regarding fluorescent aptasensor development. Generally, electrochemical aptasensors are rapid, easy, and higher in sensitivity compared to optical sensors, making them the best candidates for the on-site rapid assay of biomarkers.

**Table 1 sensors-21-00979-t001:** Examples of application of aptasensors for quantitative detection.

Sensor Type	Target	Aptamer Sequence	Detection Range	LOD	Ref.
Colorimetric	*E. coli*	Fp: TAGGGAAGAGAAGGACATATGAT, Rp: TTGACTAGTACATGACCACTTGA)	10^1^–10^8^ cells/mL	10^1^ cells/mL	[84]
	Malathion	5′-TAT ACA CAA TTG TTT TTC TCT TAA CTT CTT GAC TGC-3′	0–4000 ng/L	5.24 ng/L	[85]
	Cd^2+^	5′-CTCAGGACGACGGGTTCACAGTCCGTTGTC-3′	1–400 ng/L	1 ng/L	[86]
	*Salmonella typhimurium*	Apt1: 5′-biotin-GAGGAAAGTCTATAGCAGAGGAGATGTGTGAACCGAGTAA-3′	3.3 × 10^1^–3.3 × 10^6^ CFU/mL	33 CFU/mL	[87]
		Apt2: 5′-CTCCTCTGACTGTAACCACGGAGTTAATCAATACAAGGCGGGAACATCCTTGGCGGTGCCGCATAGGTAGTCCAGAAGCC-3′			
	*Salmonella typhimurium*	NH_2_-GCGCTCGGCCTCCTCTGCCATCTCATTCGCGAGCGC	100–10^9^ CFU/mL	16 CFU/mL	[88]
	AFM1	5-Biotin-ACTGCTAGAGATTTTCCACAT-3′	0.3–75 ng/mL	0.03 ng/mL	[89]
	AFB1	5′-GTTGGGCACGTGTTGTCTCTCTGTGTCTCGTGCCCTTCGCTAGGCCCACA-3′	1–6 ng/mL	0.18 ng/mL	[90]
	*Salmonella typhimurium*	Apt1 5′-AGT AAT GCC CGG TAG TTA TTC AAA GAT GAG TAG GAA AAG A-C_6_-SH-3′	10^1^–10^6^ CFU mL^−1^	1 CFU/mL	[97]
		Apt2 5′-TAT GGC GGC GTC ACC CGA CGG GGA CTT GAC ATT ATG ACA G-C_6_-SH-3′			
	ABA	AAAATGGGTTAGGTGGAGGTGGTTATTCCGGGAATTCGCCCTAAATACGAGCAAC	1 nM to 10 μM	0.51 nM	[98]
	Cortisol	5′-GGA ATG GAT CCA CAT CCA TGG ATG GGC AAT GCG GGG TGG AGA ATG GTT GCC GCA CTT CGG CTT CAC TGC AGA CTT GAC GAA GCT T-3′	0.1–1000 nM	0.1 nM	[99]
	Thrombin	TBA1 (5′-thiolated-TTT TTT TTT TTT TTT GGT TGG TGT GGT TGG-3′)	0–10 μg/mL	1.33 μg/mL	[100]
		TBA2 (5′-thiolated-TTT TTA GTC CGT GGT AGG GCA GGT TGG GGT GAC T-3′)			
	Cd^2+^	5′-biotin-ACC GAC CGT GCT GGA CTC TGG ACT GTT GTG GTA TTA TTT TTG GTT GTG CAG TAT GAG CGA GCG TTG CG-3	1–500 ng/mL	0.7 ng/mL	[101]
	PDGF-BB	5′-CAGGCTACGGCACGTAGAGCATCACCATGATCCTG-3′	1–25 pM	0.94 pM	[103]
	ATP	5′-ACC TGG GGG AGT ATT GCG GAG GAA GGT-3′	0.50–100 μM	0.09 μM	[106]
	Pb^2+^	5′-biotin-GGGTGGGTGGGTGGGT-3′	1–300 ng/mL	0.63 ng/mL	[107]
	*E. coli*	Apt1: 5′-biotin-TGAGCCCAAGCCCTGGTATGCGGATAACGAGGTATTCACGACTGGTCGTCAGGTATGGTTGGCAGGTCTACTTTGGGATC-3′	16 to 1.6 × 10^6^ CFU/mL	2 CFU/mL	[108]
		Apt1: 5′-biotin-TGAGCCCAAGCCCTGGTATGAGCCCACGGAACACTGGTCGCGCCCACTGGTTTCTATATTGGCAGGTCTACTTTGGGATC-3′			
	17β-E2	5′-GCTTCCAGCTTATTGAATTACACGCAGAGGGTAGCGGCTCTGCGCATTCAATTGCTGCGCGCTGAAGCGCGGAAGC-3′	1.5–50 nM	1.5 nM	[109]
	AFB1	5′-biotin-GTT GGG CAC GTG TTG TCT CTC TGT GTC TCG TGC CCT TCG CTA GGC CCA CA-3′	ND	0.1 ng/mL	[110]
	PSA	5′-Biotin-ATTAAAGCTCGCCATCAAATAGC-3′	0–2.1 ng/mL	0.15 ng/mL	[111]
	*E.coli O157: H7*	5′-ATCCGTCACACCTGCTCTGTCTGCGAGCGGGGCG	500–5 × 10^7^ CFU/mL	250 CFU/mL	[112]
		CGGGCCCGGCGGGGGATGCGTGGTGTTGGCTCCCGTAT-3′			
Fluorometric	T-2	5′-CAGCTCAGAAGCTTGATCCTGTATATCAAGCATCGCGTGTTTACACATGCGAGAGGTGAAGACTCGAAGTCGTGCATCTG-3′	0.001−100 ng/mL	0.57 pg/mL	[119]
	DGX	5′-AGCGAGGGCGGTGTCCAACAGCGGTTTTTTCACGAGGAGGTTGGCGGTGG-3′	0.001 to 0.5 ng/mL	0.0032 ng/mL	[120]
	ZEN	5′-NH_2_-AGCAGCACAGAGGTCAGATGTCATCTATCTATGGTACATTACTATCTGTAATGTGATATGCCTATGCGTGCTACCGTGAA-3′	31.4–628 nM	7.5 nM	[121]
	PAT	5′-GGC CCG CCA ACC CGC ATC ATC TAC ACT GAT ATT TTA CCT T-3′CFL	5–300 ng/mL	0.13 ng/mL	[130]
	AFB1	TARMA-5′-GTT GGG CAC GTG TTG TCT CTC TGT GTC TCG TGC CCT TCG CTA GGC CCA CA-3′	0–180 ng/mL	0.35 ng/mL	[123]
	AMP	5′-CACGGCATGGTGGGCGTCGTG-Thiol-3′	100–1000 pM	29.2 pM	[131]
	IFN-γ	Apt1: 5′-H_2_N-C_6_-CCGCCCAAATCCCTAAGAGAAGACTGTAATGAC ATCAAACCAGACACACACTACACACGCA-3′	2.0 × 10^−18^–5.0 × 10^−8^ M	0.178 fM	[132]
		Apt2: 5′-TGGGGTTGGTTGTGTTGGGTGTTGTG-Azide(N_3_)-3′			
	MUC1	5′-Cy3-GCAGTTGATCCTTTGGATACCCTGG-NH_2_-3′	0–50 nM	0.8 nM	[133]
	Isocarbophos	5′-AGCT_2_GCTGCAGCGAT_2_CT_2_GATCGC_2_ACAGAGCT-3’	10–500 nM	3.38 nM	[134]
	Hg^2+^	5′-FAM-TTC TTT CTT CCC CTT GTT TGT T-3′	20–150 nM	15 nM	[135]
	*E. coli*	5′-CCG GAC GCT TAT GCC TTG CCA TCT ACA GAG CAG GTG TGA CGG-C_6_ NH_2_-3′	85 to 85 × 10^7^ CFU/mL	17 CFU/mL	[140]
	*E. coli* ATCC8739	5′-ATCCGTCACACCTGCTCTGTCTGCGAGCGGGGCGCGGGCCCGGCGGGGGATGCGTGGTGTTGGCTCCCGTAT-3′	58–58 × 10^6^ CFU/mL	10 CFU/mL	[141]
	Malathion	5′-ATCCGTCACACCTGCTCTTATACACAATTGTTTTTCTCTTAACT TCTTGACTGCTGGTGTTGGCTCCCGTAT-3′	of 0.01–1 μM	1.42 nM	[142]
	Pb^2+^	Apt1: 5′-Biotin-CGA TCA CTA ACT ATr AGG AAG AGA TG-HS-3′	25–1400 nM	5.7 nM	[143]
		Apt2: 5′-NH_2_-TGA GTG ATA AAG CTG GCC GAG CCT CTT CTC TAC-3′			
	Chlorpyrifos	5′-CCTGCCACGCTCCGCAAGCTTAGGGTTACGCCTGCAGCGATTCTTGATCGCGCTGCTGGTAATCCTTCTTTAAGCTTGGCACCCGCATCGT-3′	5–600 nM	3.8 nM	[144]
	ATP	5′-CCCCAACTCCTTCCCGAAACCTACCTGGGGGAGTATTGCGGAGGAAGGTTTCGGG-3′	20–220 μM	0.38 μM	[145]
	ATP	5′-CCCCCCCCCCCCCACCTGGGGGAGTATTGCGGAGGAAGGT-3′	50 pM–1.0 nM	26 pM	[152]
	AFB1	5′-GTT GGG CAC GTG TTG TCT CTC TGT GTC TCG TGC CCT TCG CTA GGC CCA CA-3′	2–400 ng/mL	1.82 ng/mL	[153]
	Acetamiprid	5′-TGT AAT TTG TCT GCA GCG GTT CTT GAT CGC TGA CAC CAT ATT ATG AAG A-3′	5 nM–1.2 μM	3 nM	[154]
	Exosomes	5′-CATCCATGGGAATTCGTCGACCCTGCAGGCATGCAAGCTTTCCCTATAGTGAGTCGTATTACTGCCTAGGCTCGAGCTCG-3′	0.68–30.4 pM	0.57 pM	[121]
	ATP	5′-FAM-AATTCTGGGGGAGCCTTTTGT GGG TAGGGC GGG TTG GTT TTG CCC CGG AGG AGG AATT-BHQ1-3′	1−500,000 μM	ND	[159]
	Cd^2+^	5′-AGTGACGTGCTGGACTCCGGACTATTGTGGTATGATCTGGTTGTGACTATGCAGTGCGTGCA-(CH_2_)_3_-SH-3′	20 pM–12 nM	1.2 pM	[161]
	MC-LR	5′-GGC GCC AAA CAG GAC CAC CAT GAC AAT TAC CCA TAC CAC CTC ATT ATG CCC CAT CTC CGC-3′	0.01 to 1000 μg/L	4.8 ng/L	[162]
	Chloramphenicol	5′-AGCAGCACAGAGGTCAGATGACTTCAGTGAGTTGTCCCACGGTCGGCGAGTCGGTGGTAGCCTATGCGTGCTACCGTGAA-3′	10–107 pg/mL	0.987 pg/mL	[163]
Electrochemical	OTA	5′-triple SH-GAT CGG GTG TGG GTG GCG TAA AGG GAG CAT CGG ACA-3′	1 × 10^−5^–10 nM	3.3 × 10^−3^ pM	[190]
	Amoxicillin	5′-(SH)-TTA GTT GGG GTT CAG TTG G-3′	0.5–3 nM	0.2 nM	[191]
	Thrombin	Apt1: 5′-COOH-(CH_2_)_10_-GGTTGGTGTGGTTGG-3′.	1 pM–10 nM	0.64 pM	[192]
		Apt2: 5′-NH_2_-(CH_2_)_6_-AGTCCGTGGTAGGGCAGGTTGGGGTGACT-3′			
	PSA	5′-NH_2_-TTT TTA ATT AAA GCT CGC CAT CAA ATA GCT TT-3′	0.0001–100 ng/mL	0.085 pg/mL	[169]
	OTC	5′-SH-GGA ATT CGC TAG CAC GTT GAC GCT GGT GCC CGG TTG TGG TGC GAG TGT TGT GTG GAT CCG AGC TCC ACG TG-3′	1 × 10^−13^ to 1 × 10^−5^ g/mL	3.1 × 10^−14^ g/mL	[196]
	PAT	5′-NH_2_-GGCCCGCCAACCCGCATCATCTACACTGATATTTTACCTT-3′	5 × 10^−8^–5 × 10^−1^ μg/	1.46 × 10^−8^ μg/mL	[43]
	Thrombin	5′-AGTCCGTGGTAGGGCAGGTTGGGGTGACT-3′	1 fM–1 nM	0.57 fM	[13]
	Pb^2+^	5′-GGGTGGGTGGGTGGGT-3′ and its complementary strand 5′-CCACCCACCC–(CH_2_)_6_–SH-3′	0.5–25 ppb	0.6 ppb	[197]
	H5N1	5′-biotin-GTGTGCATGGATAGCACGTAACGGTGTAGTAGTAACGTGCGGGTAGGAAGAAAGGGAAATAGTTGTCGTGTTG-3′	0.001 to 1 HAU	8 × 10^–4^ HAU	[201]
	Sulfadimethoxine	5′-GAG GGC AAC GAG TGT TTA TAG A-3′, DNA probe, 5′–SH–TCT ATA AAC ACT CGT TGC CCT C-3′	0.1–500 nM	0.038 nM	[203]
	Malathion	5′-COOH-ATCCGTCACACCTGCTCTTATACACAATTGTTTTTCTCTT AACTTCTTGACTGCTGGTGTTGGCT-3′	0.5–600 ng/L	0.5 ng/L	[204]
	ATZ	5′-TGT-ACC-GTC-TGA-GCG-ATT-CGT-ACG-AAC-GGC-TTT-GTA-CTG-TTT-GCA-CTG-GCG-GAT-TTA-GCC-AGT-CAG-TGT-TAA-GGA-GTG-C-3′	50-fM–0.3 nM	12.0 fM	[206]
	OTA	5′-AAAGATCGGGTGTGGGTGGCGTAA AGGGAGCATCGGACA-3′	0.01–1 ng/mL	0.004 ng/mL	[205]

Biomarker detection based on aptamer functionalization still needs to overcome these limitations in order to be available for the multi-detection of metal ions, DNA, and proteins. Furthermore, the development of more specific aptamers is still needed, as is also integration into sensor platforms. Aptasensor platforms need more attention regarding (1) simultaneous multiple marker detection; (2) the long-term stability of biosensor assays; (3) direct assays in real sample matrixes; (4) understanding the nature of the binding competition between an aptamer and target on the surface of a nanomaterial, which could affect the sensor reliability; (5) more focus on the future development of in vivo aptamer sensing technology, the possible problems, and their solutions; and (6) more intensive research regarding the improvement of the POCT of biomarkers using aptasensors. The aforementioned hurdles need to be overcome quickly for the achievement of a reliable and selective detection of markers with ultrasensitivity, with affordable and portable on-site analytical devices. We are sure that the scientific community has the talent to offer solutions to these hurdles in order to design an affordable, easy-to-use nanomaterial-based electro-optical aptasensor integrated with rolling-cycle amplification technology.

## Figures and Tables

**Figure 1 sensors-21-00979-f001:**
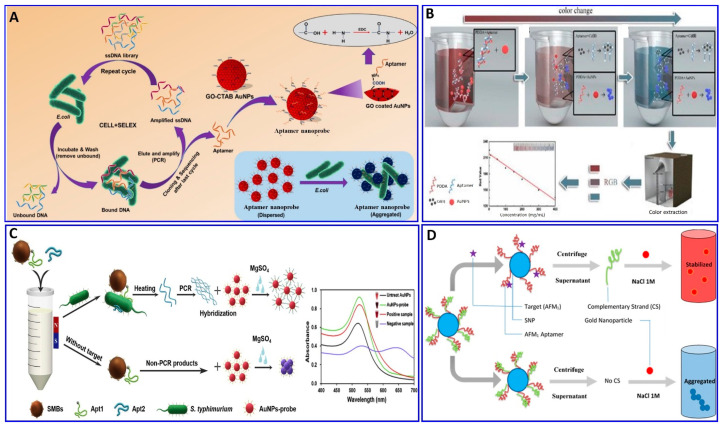
(**A**) Schematic illustration of the colorimetric aptamer sensor for *E. coli* assay via target inducing aggregation of gold nanoparticles (AuNPs), reproduced from [84]. (**B**) Schematic illustration of the colorimetric aptasensor for Cd^2+^ assay based on AuNP aggregation, reproduced from [86]. (**C**) Schematic procedure for an *S. typhimurium* bacterium assay via a proposed colorimetric SMBs-Apt1 sandwich strategy, reproduced from [87]. (**D**) Schematic illustration of a colorimetric aptasensor assay for AFM1 based on salt aggregation of AuNPs induced by AuNP–aptamer competitive binding, reproduced from [89].

**Figure 3 sensors-21-00979-f003:**
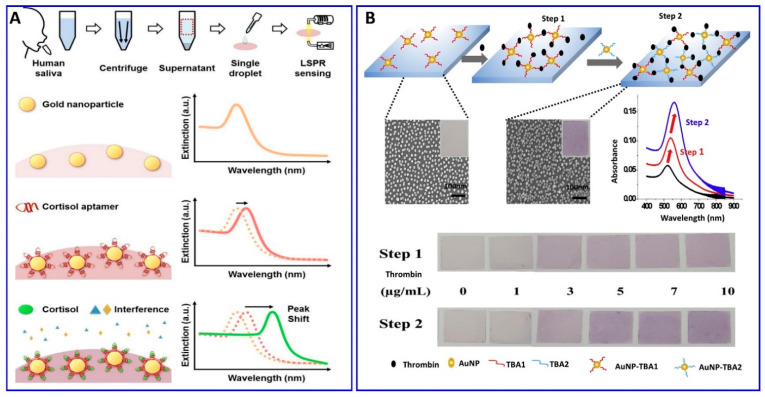
(**A**) Schematic illustration of the cortisol assay based on localized surface plasmon resonance (LSPR) aptasensor, reproduced from [99]. (**B**) Scheme representing the thrombin assay via sandwich colorimetric solid-phase aptasensor based on the enhanced LSPR, reproduced from [100].

**Figure 5 sensors-21-00979-f005:**
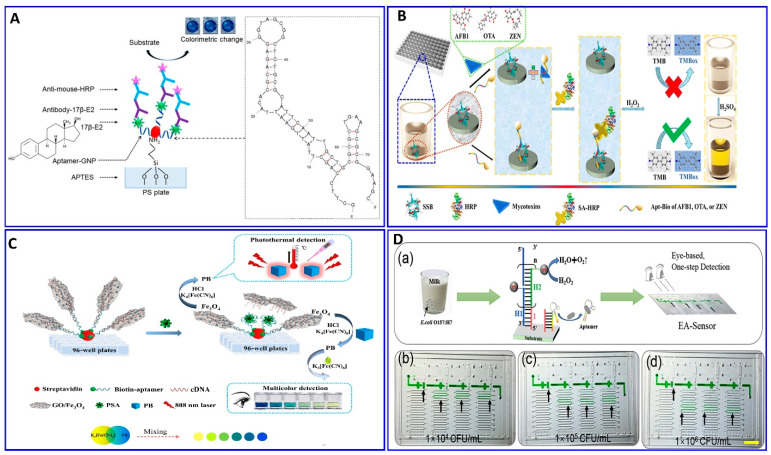
(**A**) Schematic illustration of the ELISA for assaying of 17β-E2 with an assisting aptamer–antibody sandwich functionalized with AuNPs, reproduced from [109]. (**B**) Principal assay procedures for mycotoxins using a green ELISA based on a single-stranded binding protein (SSB)-assisted aptamer; the SSB was dispensed into polystyrene 96 plates; reproduced from [110]. (**C**) Schematic illustration outlining the multicolor and photothermal assay for prostate-specific antigen (PSA) via magnetic beads assisting aptamer separation, reproduced from [111]. (**D**) Detection of *E.coli O157:H7* in milk via the developed naked-eye EA-Sensor, reproduced from [112].

**Figure 9 sensors-21-00979-f009:**
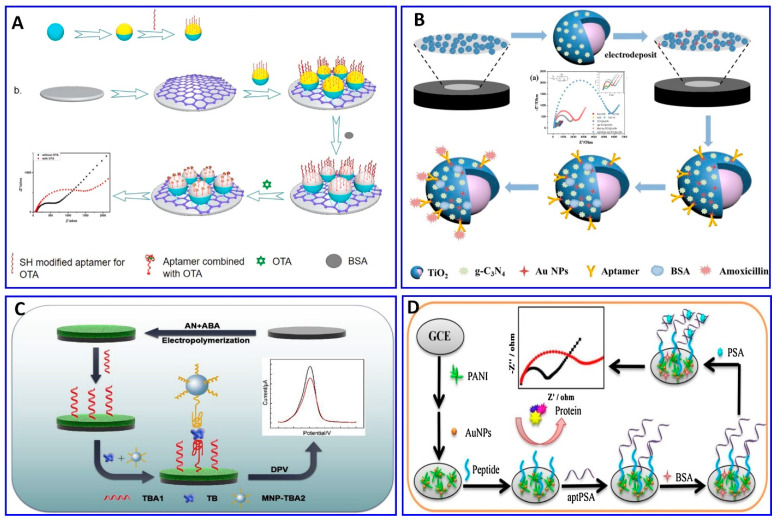
(**A**–**D**) Schematic illustrations for the fabrication of electrochemical aptasensors. (**A**) Impedimetric assay using glassy carbon electrode (GCE) immobilized with OTA aptamer, reproduced from [190]. (**B**) Impedimetric assay for amoxicillin based on the synergetic effect between the amoxicillin and its aptamer on the surface of the GCE coated with TiO_2_-g-C_3_N_4_-AuNPs, reproduced from [191]. (**C**) A sandwich-type aptasensor for thrombin assay using CSPH hydrogel-modified electrode, reproduced from [192]. (**D**) PSA assay based on polyaniline (PANI)/AuNPs as a conducting layer on GCE immobilized with a PSA aptamer, reproduced from [193].

**Figure 10 sensors-21-00979-f010:**
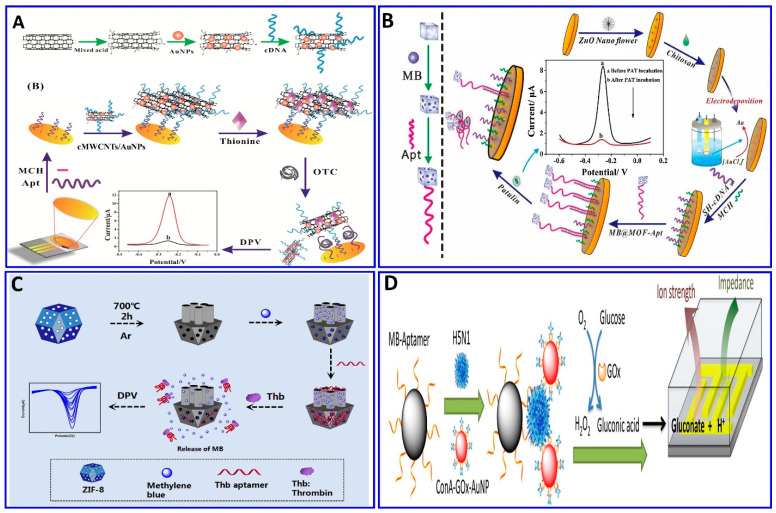
(**A**–**C**) Schematic protocols for electrochemical aptasensor based on the target-responsive label electroactive dye amplification strategy. (**A**) OTC assay using AuNP/cMWCNT/cDNA@thionine probe as a signaler releasing thionine, reproduced from [196]. (**B**) PAT assay using MOF@Methylene blue signal tags, with signals as a function of PAT concentrations, reproduced from [43]. (**C**) Thrombin assay based on target-responsive methylene blue (MB) release from the ZIF-8 surface as a function of the thrombin concentration, reproduced from [13]. (**D**) Schematic protocols for an electrochemical aptasensor based on an enzymatic catalytic reaction, reproduced from [201].

**Figure 11 sensors-21-00979-f011:**
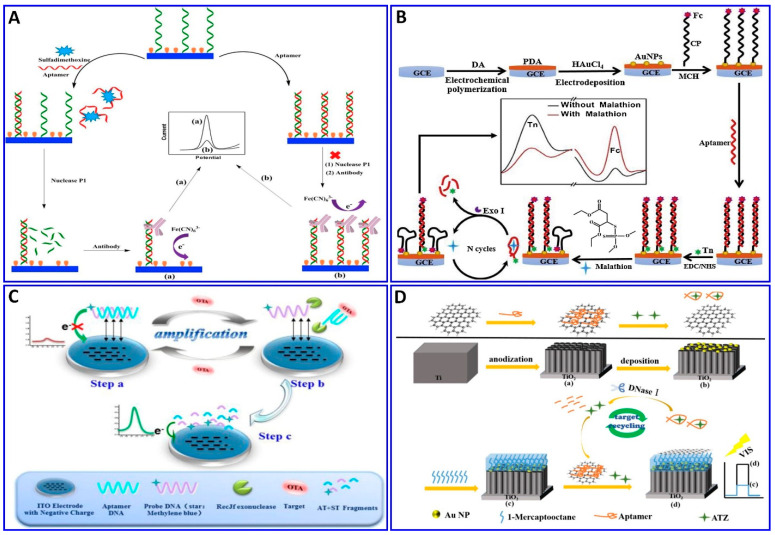
(**A**–**C**) Schematic protocols for electrochemical aptasensor integrated with signal amplification based on nuclease strategy. (**A**) Assaying of sulfadimethoxine using gold electrode fabricated with dsDNA consisting of DNA probe and specific aptamer for sulfadimethoxine inhibiting nuclease digestion, reproduced from [203]. (**B**) Malathion assay with the assistance of Exo I as a signal enhancer and layer of PDA-AuNPs deposited on GCE as an electrical conductivity enhancer, reproduced from [204]. (**C**) OTA assay using TiO_2_ nanotube electrode functionalized with DNA probe tagged with MB, reproduced from [205]. (**D**) Schematic protocols for photoelectrochemical aptasensor ATZ assay with nuclease signal amplification using TiO_2_ nanotube covered by a layer of AuNPs, further functionalized with ATZ aptamer/graphene mixture as a sensor probe, reproduced from [206].

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
