# Peer review of "Recent Advances in Aptamer Sensors"

_sensors, 2021, doi:10.3390/s21030979_

Round 1
Reviewer 1 Report
Comments are in the docx.

Reviewer 2 Report
Dear Authors,
Please find my comments inserted within the manuscript pdf document attached.

Reviewer 3 Report
This review article summarizes the literature associated with the design of aptamer-based colorimetric, fluorescent, and electrochemical sensors. The authors provide a detailed discussion that includes integrating nanomaterials with aptamers and their sensing mechanism and quantification performance. It appears valuable for the community in the field of aptamer-based sensors. However, this review article still has room to be improved:
(1) The authors did not discuss aptamer-based sensors' practical applications in real-world samples.
(2) The authors did not describe the advantages and disadvantages of each material in combination with aptamer.
(3) Please build up the table to record the information shown in this review article.
(4) The authors did not include aptamer-related SERS sensors in this review article.
Since this well-written manuscript fits this Journal's scope, I recommend the publication of this review article after a minor revision has been made.
Reviewer 4 Report
Please see attachment.

Round 2
Reviewer 4 Report
The authors have made a large effort for the improvement of the manuscript and all the concerns raised in the first evaluation have been taken care with excellence. I do recommend the publication of the review in the present form.